# Impact of Liver Inflammation on Bile Acid Side Chain Shortening and Amidation

**DOI:** 10.3390/cells11243983

**Published:** 2022-12-09

**Authors:** Marta Alonso-Peña, Ricardo Espinosa-Escudero, Heike M. Hermanns, Oscar Briz, Jose M. Herranz, Carmen Garcia-Ruiz, Jose C. Fernandez-Checa, Javier Juamperez, Matias Avila, Josepmaria Argemi, Ramon Bataller, Javier Crespo, Maria J. Monte, Andreas Geier, Elisa Herraez, Jose J. G. Marin

**Affiliations:** 1Experimental Hepatology and Drug Targeting (HEVEPHARM), Institute for Biomedical Research of Salamanca (IBSAL), University of Salamanca, 37007 Salamanca, Spain; 2Gastroenterology and Hepatology Department, Clinical and Translational Research in Digestive Diseases, Valdecilla Research Institute (IDIVAL), Marqués de Valdecilla University Hospital, 39008 Santander, Spain; 3Division of Hepatology, Würzburg University Hospital, Medical Clinic II, 97080 Würzburg, Germany; 4Centro de Investigación Biomédica en Red de Enfermedades Hepáticas y Digestivas (CIBERehd), Carlos III National Institute of Health, 28029 Madrid, Spain; 5Hepatology Program, Liver Unit, Instituto de Investigación de Navarra (IdisNA), Clínica Universidad de Navarra and Centro de Investigación Médica Aplicada (CIMA), Universidad de Navarra, 31008 Pamplona, Spain; 6Institute of Biomedical Research of Barcelona (IIBB), Centro Superior de Investigaciones Científicas (CSIC), 08036 Barcelona, Spain; 7Center for ALPD, Keck School of Medicine, University of Southern California, Los Angeles, CA 90089, USA; 8Pediatric Hepatology and Liver Transplantation Unit, Vall d’Hebron University Hospital (HVH), Universitat Autónoma de Barcelona, 08035 Barcelona, Spain; 9Center for Liver Diseases, Pittsburgh Liver Research Center, Division of Gastroenterology, Hepatology and Nutrition, University of Pittsburgh Medical Center, Pittsburgh, PA 15213, USA

**Keywords:** ACOX2, ASH, BAAT, bile acid, inflammation, NAFL, NASH, oncostatin M

## Abstract

Bile acid (BA) synthesis from cholesterol by hepatocytes is inhibited by inflammatory cytokines. Whether liver inflammation also affects BA side chain shortening and conjugation was investigated. In human liver cell lines (IHH, HepG2, and HepaRG), agonists of nuclear receptors including the farnesoid X receptor (FXR), liver X receptor (LXR), and peroxisome proliferator-activated receptors (PPARs) did not affect the expression of BA-related peroxisomal enzymes. In contrast, hepatocyte nuclear factor 4α (HNF4α) inhibition down-regulated acyl-CoA oxidase 2 (ACOX2). *ACOX2* was repressed by fibroblast growth factor 19 (FGF19), which was prevented by extracellular signal-regulated kinase (ERK) pathway inhibition. These changes were paralleled by altered BA synthesis (HPLC-MS/MS). Cytokines able to down-regulate cholesterol-7α-hydroxylase (CYP7A1) had little effect on peroxisomal enzymes involved in BA synthesis except for ACOX2 and bile acid-CoA:amino acid N-acyltransferase (BAAT), which were down-regulated, mainly by oncostatin M (OSM). This effect was prevented by Janus kinase (JAK) inhibition, which restored BA side chain shortening and conjugation. The binding of OSM to the extracellular matrix accounted for a persistent effect after culture medium replacement. In silico analysis of four databases (*n* = 201) and a validation cohort (*n* = 90) revealed an inverse relationship between liver inflammation and ACOX2/BAAT expression which was associated with changes in HNF4α levels. In conclusion, BA side chain shortening and conjugation are inhibited by inflammatory effectors. However, other mechanisms involved in BA homeostasis counterbalance any significant impact on the serum BA profile.

## 1. Introduction

In non-alcoholic steatohepatitis (NASH), elevated serum bile acid (BA) concentrations (mainly conjugated forms) have been correlated with histological features of the disease [1]. Whether this alteration in BA homeostasis is involved in the pathogenesis of NASH or is a consequence of the inflammatory process remains unknown [2].

Besides the well-known role of BAs in the digestive function as fat emulsifiers, they are also involved in the transcriptional control of proteins involved in their synthesis, metabolism, and transport [3] by interacting with plasma membrane receptors, such as G-protein-coupled BA receptor 1 (GPBAR1 or TGR5) and nuclear receptors, such as the farnesoid X receptor (FXR), pregnane X receptor (PXR) and vitamin D receptor (VDR). BAs are also metabolic integrators involved in the management of fat, sugar, and energy metabolism [4], affecting stellate cells proliferation [5] and modulating the participation of Kupffer cells in liver inflammation [6].

BAs are synthesized by hepatocytes through several metabolic pathways, i.e., the main classic/neutral route, the alternative acidic pathway, and at least two other minor routes. The first reaction in the classic pathway is the conversion of cholesterol into 7α-hydroxycholesterol, which is the rate-limiting step of BA synthesis in adult humans. Therefore, the overall regulation of BA synthesis has been linked to the control of the enzyme that catalyzes this reaction, CYP7A1. 7α-hydroxycholesterol is efficiently transformed into 7α-hydroxy-4-cholesten-3-one (C4), whose serum levels are used as an indirect biomarker of CYP7A1 activity. In rodents, Cyp7a1 expression is under the control of three main nuclear receptors, i.e., PPARα [7], LXR [8], and FXR [9]. In humans, intestinal and hepatic FXR play crucial roles through the up-regulation of FGF19 and the small heterodimer partner (SHP), respectively [10]. FGF19 activates hepatocyte membrane receptor FGFR4/β-Klotho, which inhibits *CYP7A1* expression through the activation of the ERK pathway [11].

In hepatocytes, FXR activation up-regulates SHP, which acts as a co-repressor that competes with peroxisome proliferator-activated receptor gamma coactivator 1 alpha (PGC1). This favors the expression of genes involved in BA synthesis [3]. Moreover, SHP interacts with several transcription factors, such as hepatocyte nuclear factor 4α (HNF4α, gene *HNF4A* or *NR2A1*), behaving as a dominant negative protein also inhibiting their transcriptional effect on genes involved in BA synthesis, such as CYP7A1 and CYP8B1 [12,13]. Moreover, HNF1α is also an essential transcriptional regulator of BA metabolism through the control of FXR and HNF4α expression [14,15]. MafG has been identified in mice as an FXR target gene acting as a transcriptional repressor of genes involved in BA synthesis [16].

In addition, the liver plays a peculiar immunological role due to its enriched resident immune cell population controlling the balance between immune tolerance and response [17]. In the pathophysiology of liver injury, the interaction of immune cells, hepatocytes, cholangiocytes, and other resident cells plays an important role. In this context, the inhibitory effect of several cytokines on the expression of enzymes such as tumor necrosis factor α (TNFα) and interleukin 1β (IL1β), (human CYP7A1) [18] and transporters (rat Ntcp, Oatps, Mrp2, Mrp3, and Bsep) involved in BA homeostasis has been described [19,20,21]. Thus, TNFα and IL1β, upon binding to different receptors, activate MEK4/7 and JNK1/2 pathways, which induces HNF4α phosphorylation. Since this change reduces HNF4α transcriptional activity [22], there is a subsequent inhibition of *CYP7A1* transcription [18]. In addition, transforming growth factor β1 (TGFβ1) activates SMAD3 and JNK pathways in hepatocytes, impairing HNF4α activity and therefore reducing CYP7A1 expression [23]. Hepatocyte growth factor (HGF) also reduces *CYP7A1* transcription in hepatocytes acting via SHP, cJun, ERK1/2, and JNK [24].

Peroxisomes play a crucial role in BA synthesis [25]. After the initial modification of the cholesterol sterol ring, the C27-BA intermediates 3α,7α-dihydroxy-5β-cholestanoic acid (DHCA) and 3α,7α,12α-trihydroxy-5β-cholestanoic acid (THCA) undergo β-oxidation of their side chain to generate the C24 primary BAs chenodeoxycholic acid (CDCA) and cholic acid (CA), respectively. C27-BA intermediates are translocated across the peroxisome membrane by the ATP binding cassette transporter ABCD3. Their subsequent β-oxidation is catalyzed by α-methyl-acyl-CoA racemase (AMACR), acyl-CoA oxidase 2 (ACOX2), D-bifunctional protein (DBP), and sterol carrier protein X (SCPx) [25].

ACOX2 deficiency-associated hypertransaminasemia (ADAH) has been recently described in several patients with unexplained biochemical signs of hepatocellular damage [26,27,28]. This condition can be accurately identified by a noninvasive diagnostic strategy based on plasma BA profiling, characterized by the predominance of C27-BA intermediates, and ACOX2 sequencing. Ursodeoxycholic acid (UDCA) treatment seems to efficiently attenuate liver damage in these patients [29]. Whether there is a link between ACOX2 expression and liver inflammation is unknown.

Under physiological conditions, less than 2% of the BAs present in human bile are in their unconjugated form. The peroxisomal enzyme BA-CoA:amino acid N-acyltransferase (BAAT) catalyzes the amidation with taurine or glycine of BAs which can be either synthesized de novo or recycled back to the liver via the enterohepatic circulation after being deconjugated by intestinal microbiota [30,31]. Conjugation plays a crucial physiological role because the conjugated species are better substrates for BA transporters accounting for their efficient enterohepatic circulation. Accordingly, BAAT deficiency results in the intrahepatic accumulation of unconjugated BAs [32]. Conjugation decreases BA hydrophobicity, which favors micelle-forming and hence fat digestion and the intestinal absorption of lipids and lipid-soluble vitamins. Thus, impaired conjugation could result in malabsorption [31]. The mechanisms involved in the transcriptional control of BAAT and its response to inflammation are poorly understood.

Here we aimed to investigate the impact of inflammatory processes occurring in many liver diseases, some of them with a high prevalence like NASH, on the expression of peroxisomal enzymes involved in the shortening and conjugation of BA side chains and their potential relationship with the altered BA homeostasis described in these patients [1,2].

## 2. Materials and Methods

### 2.1. Ethics

Research protocols used in the study of serum BAs were approved by the Würzburg University Hospital Ethical Committee for Clinical Research, Würzburg, Germany (ref. 96/12 and 188/17). All patients signed written consent forms for the use of their samples for biomedical research. Regarding RNAseq data of the validation cohort, this was an in silico re-analysis of the work published elsewhere [33], where all information concerning the institutional review board is available.

### 2.2. Cell Lines

HepG2 (human hepatoblastoma) cells were obtained from the American Type Culture Collection (ATCC, LGC Standards, Barcelona, Spain). Immortalized human hepatocytes (IHH) were kindly donated by Prof. Dr. F. Kuipers (Department of Pediatrics, University Hospital Groningen, Groningen, The Netherlands) and cultured according to the literature [34]. HepaRG undifferentiated cells were obtained from Gibco–Thermo Fisher (Madrid, Spain).

### 2.3. Chemicals

CA, CDCA, deoxycholic acid (DCA), lithocholic acid (LCA), ursodeoxycholic acid (UDCA), as well as their tauroconjugated (TCA, TCDCA, TDCA, TLCA, TUDCA) and glycoconjugated (GCA, GCDCA, GDCA, GLCA, GUDCA) forms, taurosulfolithocholic acid (TSLCA), and the unspecific PPARs agonist bezafibrate, were acquired from Sigma-Aldrich (Madrid, Spain). THCA, as a mixture of (25R)- and (25S)-isomers, and C4 came from Avanti Polar Lipids (Alabaster, AL, USA). SB202190, a competitive inhibitor of MAPK p38, and SP600125, a selective inhibitor of N-terminal c-Jun kinase (JNK), were purchased from Calbiochem–Sigma-Aldrich, in addition to U0126, a selective inhibitor of MEK1 and 2, from Cayman (Hamburg, Germany). Ruxolitinib, a selective inhibitor of Janus Kinase (JAK1/2), rottlerin, an inhibitor of PKCδ, BIX02189, a selective inhibitor of MEK5, fludarabine, an inhibitor of STAT1, and STAT5-IN-1, a selective inhibitor of STAT5, are from Selleckchem (Munich, Germany). The HNF4 inhibitor BI6015 was acquired from Focus Biomolecules (Plymouth Meeting, PA, USA). Human recombinant IL6, human recombinant IL1β and human recombinant TNFα were obtained from Gibco -Thermo Fisher. Human recombinant FGF19 was purchased from PeproTech (London, UK), and Oncostatin M (OSM) from PeproTech or RayBiotech (Norcross, GA, USA). The FXR agonist GW4064 was acquired from Santa Cruz Biotechnology (Santa Cruz, CA, USA). The LXR agonist GW3965 was purchased from TargetMol (Boston, MA, USA).

### 2.4. In Vitro Studies

IHH and HepG2 cells were seeded onto the appropriate plates at subconfluence (≈70,000 and 50,000 cells/cm^2^, respectively). After 16 h, the cells were treated according to the experimental design with nuclear receptors’ agonists, recombinant cytokines and inhibitors for different second messenger systems and incubated for different times before cell lysates were obtained.

ACOX2 activity was assessed by studying the conversion of THCA into CA in HepG2 cells. These cells were seeded as previously described and then incubated for 6 h with a culture medium supplemented with OSM (25 ng/mL) alone or in combination with ruxolitinib (0.5 µM), BI6015 (5 µM), or FGF19 (5 ng/mL) with or without a MEK1/2 inhibitor U0126 (5 µM). Afterward, they were incubated for an additional 48 h in culture medium containing 10 µM THCA, both with and without OSM. Finally, the culture medium was collected. Cell lysates were obtained by incubation with water at 4 °C for 4 h, scrapped and sonicated. Both medium and cell lysates were used for BAs extraction with Sep-Pak Plus C18 cartridges (Waters, Barcelona, Spain). The biotransformation of THCA into CA was analyzed by HPLC-MS/MS, as previously described [29]. The enzyme activity was corrected by culture protein content [35].

BAAT activity was studied by determining the biotransformation of CA into TCA and GCA by differentiated HepaRG cells. Before the experiment, HepaRG cells were seeded at confluence (≈100,000 cells/cm^2^) until they underwent differentiation, as previously described [36]. After that, the cells were incubated for 6 h in a culture medium supplemented with OSM (25 ng/mL), followed by an additional incubation for 48 h in a culture medium containing both CA (5 µM) and OSM. BA content in both medium and cell lysates was determined by HPLC-MS/MS [29].

### 2.5. Determination of Gene Expression

Total RNA extraction from cells and tissues and reverse transcription were performed as previously described [37,38]. cDNA synthesized from total RNA was used as a template to determine gene expression by real-time quantitative PCR (qPCR) by using gene-specific primers spanning exon-exon junctions in the target mRNA (Appendix A) and AmpliTaq Gold DNA polymerase/SYBR Green I in a 7300 Real-Time PCR System or the SYBR™Select Mastermix in a ViiA7 Real-Time PCR System (Applied Biosystems, Thermo Fisher Scientific, Madrid, Spain). The thermal cycling conditions were as follows: single cycles at 50 °C for 2 min and at 95 °C for 10 min, followed by 40 cycles at 95 °C for 15 s and at 60 °C for 60 s. The mRNA abundance in each sample was normalized based on its *HPRT* (hypoxanthine phosphoribosyltransferase) mRNA content.

To carry out *STAT3* silencing, a mix of four siRNA (On-TARGET PLUS siRNA human STAT3; ref: J-003544-7/8/9/10; 1:1:1:1) and a Mock control (On-TARGET PLUS control pool non-targeting; ref. D-001810-10-05) both supplied by Thermo Scientific Dharmacon (Horizon, Perkin Elmer, Cambridge, UK), were used for cell transfection with RNAiMAX lipofectamine (Invitrogen, Thermo Fisher Scientific).

### 2.6. Immunoblotting

Cell lysates were run in 7.5% SDS-PAGE, loading 50–100 μg of protein per lane. Primary antibodies were diluted in TBS-Tween with or without bovine serum albumin (BSA) or non-fatty milk (1–5% *p*/*v*), as described in Appendix A. Immunoreactive protein bands were visualized by ECL (Amersham Pharmacia Biotech, Amersham, UK) after incubation with appropriate secondary antibodies (IgG-HRP linked).

### 2.7. Bile Acid Measurement

An adaptation [39] of a previously described method [40] for BA measurement by HPLC-MS/MS was used in a 6420 Triple Quad LC/MS (Agilent Technologies, Santa Clara, CA, USA). The BA precursor C4 was determined in serum after acetonitrile precipitation/extraction [41] by a modification of an HPLC-MS/MS method [42]. Concentrations of the BA precursor C4, total BAs, proportion of C27-BAs, non-amidated BAs, glycine-amidated BAs, and taurine-amidated BAs were measured in serum samples collected from healthy subjects (Control, *n* = 27) and NASH patients (*n* = 29, grade F3–F4) at the University Hospital Würzburg, Germany.

### 2.8. In Silico Study

Transcriptomic high throughput data have been downloaded from the following four databases: GSE105127 [43], GSE115193 [44], GSE126848 [45] and GSE130970 [46] in fastq using SRAToolkit version 2.11.0 (https://trace.ncbi.nlm.nih.gov; accessed on 20 May 2022). In the case of microarray assays, the data were downloaded and already processed using GEOquery version 2.54.1 [47]. For RNA-sequencing, the most up-to-date state-of-the-art processing and analysis pipelines have been used [48]. Sequences were trimmed using TrimGalore version 0.6.0 with Cutadapt version 1.18 [49]. The mapping step is carried out using STAR version 020201 [50] over genome version GRCh38.p14 (https://www.ncbi.nlm.nih.gov/assembly/GCF_000001405.26/; accessed on 20 May 2022). Read counting is performed using HTseq version 0.11.0 and normalized using EdgeR version 3.28.1 [51] in R version 3.6.3 (https://www.r-project.org/; accessed on 20 May 2022). The trimmed mean of M-values (TMM) was selected as a method for normalization in addition to low expression gene filtering for differential gene expression analysis.

### 2.9. Validation Study

Taking advantage of data collected in a recently analyzed cohort [33], a validation study was carried out. The transcriptome of human liver samples from the following groups of patients was analyzed: Control (*n* = 10) from individuals with non-diseased livers; NAFL (*n* = 9), patients with non-alcoholic fatty liver; HCV (*n* = 10), patients with non-cirrhotic HCV infection; cirrhosis (*n* = 9), patients with compensated HCV-related cirrhosis; Early ASH (*n* = 12), patients with early alcoholic steatohepatitis who were non-obese with high alcohol intake and presented mild elevation of transaminases and histologic criteria of steatohepatitis; AH, patients with histologically confirmed alcoholic hepatitis suffering from either non-severe (*n* = 11) or severe (*n* = 18) condition, who were biopsied before any treatment; transplanted AH (*n* = 11), samples were collected from the liver explants in patients with AH who underwent early transplantation. Patients with malignancies were excluded from the study. The measurement of mRNA and a bioinformatic analysis was performed as described in detail elsewhere [33]. In brief, RNA extraction by phenol/chloroform separation (TRIzol, Thermox) was carried out from total RNA obtained from flash-frozen liver tissue. RNA purity and quality were assessed by automated electrophoresis (Bioanalyzer, Agilent) and was sequenced using the Illumina HiSeq2000 platform. Libraries were built using TruSeq Stranded Total RNA Ribo-Zero GOLD (Illumina). Sequencing was paired-end (2 Å~ 100 bp) and multiplexed. Ninety-four paired-end sequenced samples obtained an average of 36.9 million total reads, with 32.5 million (88%) mapped to GRCh37/hg19 human reference. A short read alignment was performed using the STAR alignment algorithm with default parameters [50]. To quantify the expression from transcriptome mappings, we employed RSEM [52].

### 2.10. Statistical Methods

Data are shown as mean ± SD. Results were statistically analyzed using the GraphPad program. For comparisons between two groups, a parametric paired *t*-test or Student’s *t*-test test were used. After ANOVA, the Bonferroni method of multiple-range testing was used to calculate the statistical significance of differences among groups. A correlation analysis was performed using the Spearman’s rank correlation coefficient (Wessa, P. 2022, Free Statistics Software, Office for Research Development and Education, version 1.2.1, URL https://www.wessa.net accessed on 20 May 2022).

## 3. Results

### 3.1. Cytokine-Induced ACOX2 and BAAT Down-Regulation

Our results confirmed a strong downregulatory effect of several inflammatory cytokines, namely TNFα, IL6, IL1β and OSM, on *CYP7A1* expression (Figure 1). This effect was absent or moderate regarding proteins involved in peroxisomal BA metabolism, except for ACOX2 and BAAT (Figure 1). OSM, a member of the IL6 cytokine family [53], was the strongest inhibitor for CYP7A1, *ACOX2* and *BAAT*, and also induced a moderate down-regulation of *ACOX1* and *ACOX3*, which are not involved in BA metabolism (Figure 1). OSM-induced *ACOX2* and *BAAT* down-regulation was fast (maximum in 6 h) and was maintained for at least 24 h (Appendix A). The same effect was observed even after removing OSM from the culture medium, which was probably due to the adherence of OSM to the extracellular matrix (Appendix A) [54].

To elucidate the mechanism of OSM interference with *ACOX2* and *BAAT* expression, several elements involved in intracellular signaling pathways potentially activated by OSM (JAK, MEK1/2, MEK5, p38 MAPK, JNK, MEK5, PKCδ, STAT1, STAT3, STAT5) were manipulated (Figure 2). The activation of OSM receptors (LIFR and OSMR) produces trans- and autophosphorylation of JAKs that recruits linker proteins accounting for signal propagation through the ERK1/2 pathway. Thus, the JAK activity’s abrogation with ruxolitinib completely blocked the OSM inhibitory effect in both HepG2 and IHH cells. In contrast, the inhibitors of p38 MAPK (SB202190), JNKs (SP600125), STAT5 (STAT5IN1), and STAT1 (fludarabine) did not abolish OSM-induced *ACOX2* and *BAAT* down-regulation (Figure 2). However, this effect was partially prevented by inhibiting PKCδ (rottlerin) and MEK1/2 (U0126), and was only moderately affected by the inhibition of MEK5 (BIX01289) (Figure 2). Although OSM strongly activates STAT3 phosphorylation, this pathway does not seem to play a role in the control of *ACOX2* and *BAAT* expression, because *STAT3* mRNA silencing in HepG2 cells (siSTAT3) resulted in a marked decrease in the amount of STAT3 protein, and consequently phosphorylation, in these cells (Appendix A), which did not significantly alter the OSM-mediated *ACOX2* (Appendix A) and BAAT repression (Appendix A). In summary, OSM seems to act on *ACOX2* expression in a STAT-independent manner, but probably through the sequence JAK/SHC + Grb2/Ras/Raf/MEK/ERK [55,56].

OSM treatment reduced the ability of HepG2 cells to synthesize CA from THCA (Figure 3A). The inhibition of this metabolic process was prevented by treatment with ruxolitinib, which was consistent with the preservation of *ACOX2* expression in the presence of OSM (Figure 2A). Moreover, OSM-induced impaired CA conjugation in differentiated HepaRG cells was also prevented by ruxolitinib (Figure 3B). Despite the prevalent BA conjugation with glycine in humans, in all these in vitro experiments carried out with human cell lines, TCA accounted for the most (>95%) conjugated BAs (TCA + GCA) (data not shown). This change was consistent with the ability of ruxolitinib to abolish OSM-induced *BAAT* down-regulation in these cells (Appendix A). Similar effects were observed in HepG2 (Figure 2B) and IHH (Figure 2D) cells.

### 3.2. Role of Major Nuclear and Membrane Receptors Controlling BA Homeostasis

The treatment of HepG2 cells with the FXR agonists GW4064 and CDCA did not significantly affect the expression of *ACOX2*, *BAAT*, and genes encoding other essential peroxisomal enzymes (AMACR, HSD17B4, and SCP2) and transporters (PMP70/ABCD3) (Figure 4). The activation of FXR was demonstrated by the induction of known FXR target genes, such as *SLC51A, NR0B2, ABCB11*, and *FGF19*, which encode the OSTα, SHP, BSEP and FGF19 proteins, respectively (Appendix A), and the repression of *CYP7A1* expression (Appendix A).

Treatment with the LXR agonist GW3965, under conditions previously described [57,58], resulted in the expected up-regulation of one of its target genes: *SREBPF1* (Appendix A). However, no stimulatory effect on *ACOX2, BAAT, AMACR, HSD17B4, SPC2,* or *ABCD3* expression was found (Figure 4).

Bezafibrate, a non-specific PPARα agonist used at an incubation time and dose based on previous reports by others [59,60] confirmed in our laboratory (data not shown), induced the expression of the target gene *ANGPTL4* (Appendix A) and moderately stimulated that of *HSD17B4*, whereas *ACOX2* and *BAAT* expression were unaffected (Figure 4).

We confirmed that the incubation of HepG2 cells with the HNF4α inhibitor BI6015 resulted in a marked inhibition of *CYP7A1* expression (Appendix A). Furthermore, *ACOX2* and *AMACR* were down-regulated, whereas BAAT and *HSD17B4* were up-regulated (Figure 4). Consistently, the ability of these cells to biotransform exogenous THCA into CA (Figure 3C) and endogenous DHCA into CDCA (Figure 3D) was impaired.

The incubation of HepG2 cells with recombinant human FGF19 moderately decreased *ACOX2* expression but did not significantly change the expression of other enzymes assayed. In contrast, when the specific MEK1/2 inhibitor U0126 was combined with FGF19, the expression of several enzymes, including *ACOX2*, was stimulated (Figure 4). Interestingly, U0126 was not able to restore FGF19-induced decreased biotransformation of exogenous THCA into CA (Figure 3E), whereas the endogenous conversion of DHCA into CDCA was inhibited by FGF19 and completely restored by U0126 (Figure 3F).

### 3.3. Impact of Liver Inflammation on ACOX2 and BAAT Expression

In silico studies revealed a clear trend toward the down-regulation of *ACOX2* in NASH patients included in four datasets. The effect was absent or less marked in NAFL (Figure 5A–D), suggesting that inflammation is an important differential event in the control of *ACOX2* expression. Results from the validation cohort supported a decrease in *ACOX2* expression linked to the inflammatory process (Figure 5E). The investigational study suggested that *BAAT* expression changed in the same manner as that of *ACOX2* (Figure 5F–I), which was confirmed by data obtained in the validation cohort (Figure 5J). Although *OSM* expression was feeble in all groups, even below the detection threshold in some cases, there was also a trend to be enhanced in NASH compared with Control and NAFL (Figure 6A–D), which was confirmed in the validation cohort (Figure 6E). In the investigational cohorts, there was a trend to moderately decreased *HNF4α* expression associated with the inflammatory process (Figure 6F–I). This effect was not clearly seen in the validation cohort, where *HNF4α* expression was down-regulated in both NAFL and NASH (Figure 6J).

The study of the relative change versus Control of two genes in parallel in each group of patients revealed that in both investigational and validation cohorts, there were significant correlations between the changes in *ACOX2* and *HNF4α* expression (Figure 7A,B), as well as between those of *BAAT* and *HNF4α* (Figure 7C,D). Consistently, a positive correlation between changes in the expression of *ACOX2* and *BAAT* was found (Figure 7E,F). The low expression of OSM and the lack of detection in some samples precluded carrying out an accurate similar analysis. Interestingly, these inflammatory-associated changes did not parallel changes in the expression of CYP7A1 in these patients (Appendix A).

The analysis of serum C4 levels and BA profiles in healthy individuals and patients with NASH revealed that BA synthesis through the classical pathway was activated in NASH, as indicated by fourfold enhanced C4 levels (Figure 8A), which was also consistent with fourfold enhanced levels of total BAs (Figure 8B). Despite NASH-associated *ACOX2* and *BAAT* down-regulation, alteration in the proportion of C27-BA intermediates in serum was not found (Figure 8C), and the ratio of conjugated versus non-conjugated BAs was not decreased but, on the contrary, was increased in NASH patients (Figure 8D–F).

## 4. Discussion

Despite the advances in our understanding of BA pathophysiology, their role in liver inflammation, such as that occurring in NASH, is still poorly understood [61]. Our results have shown that enzymes such as ACOX2 and BAAT involved in BA side chain shortening and conjugation are markedly down-regulated by inflammatory cytokines, mainly OSM (Figure 9).

In human peroxisomes, there are three acyl-CoA oxidases that differ in their substrate specificity, i.e., ACOX1 (very long-chain fatty acids, dicarboxylic acids and poly-unsaturated fatty acids), ACOX2 (branched-chain fatty acids and C27-BA intermediates), and ACOX3 (branched-chain fatty acids, but not C27-BAs) [28]. In contrast to ACOX1, which is a PPARα target gene [62], our results revealed that this is not the case for ACOX2. Furthermore, the control of ACOX2 expression does not involve other nuclear receptors, such as FXR and LXR. Instead, we confirmed the regulatory role of HNF4α, which is mainly expressed in liver tissue, where it plays an essential role during fetal development, cell differentiation, metabolism, and regeneration [63]. Although we have identified a predicted binding site for this transcription factor in the promoter region of *ACOX2* (data not shown), a direct interaction of HNF4α with this sequence has not been demonstrated yet.

The existence of a positive correlation between *ACOX2* and *HNF4A* expression in inflammatory liver diseases supports the hypothesis that *ACOX2* expression can be controlled by HNF4α, which is consistent with the predominant ACOX2 expression in hepatocytes [28]. Defective HNF4α regulation has been recently described as a driver of hepatocellular failure in alcoholic hepatitis [33]. The transcriptomic analysis of liver tissue from patients with early ASH, non-severe and severe AH, and patients who needed an emergency liver transplant due to AH revealed profound gene expression reprogramming, with down-regulation of HNF4A and other liver-specific transcription factors. Among the enzymes involved in liver metabolism affected by these changes, linear and progressive decreased ACOX2 expression along with disease stage has been reported [33]. The present study further supports HNF4α-dependent *ACOX2* expression, which could also be the case of BAAT, although in vitro results, obtained in HepG2 cells, were not consistent with this possibility.

To study the impact of Hnf4α depletion in adult mouse liver in the absence of the severe hepatic dysfunction leading to early mortality that occurs in animals with constitutive silencing of the gene *Hnf4a*/*Nr2a1*, a temporal, hepatocyte-specific Hnf4α knock-out mouse (Hnf4α*^F/F;AlbERT2cre^*) was generated using inducible Cre-ERT recombinase technology by Bonzo et al. [64]. In this animal model, hepatic Hnf4α was efficiently silenced by exposure to tamoxifen and transcriptome analyses were performed by microarray in mice liver. Our in silico analysis of publicly available expression data from that study (GEO accession number GSE34581) revealed that tamoxifen-induced Hnf4α depletion causes the significant (*p* < 0.05) down-regulation of both *Acox2* (−30%) and *Baat* (−51%) mRNA (data not shown).

FGF19 is involved in the regulation of *ACOX2*, but not *BAAT* expression. Treatment of liver cells with FGF19 resulted in decreased *ACOX2* expression and subsequent overall enzymatic biotransformation of exogenous THCA, which was mediated by the activation of the MEK-ERK1/2 pathway. Furthermore, CDCA biosynthesis from endogenous DHCA was also reduced by FGF19. This effect was completely prevented by inhibiting the MEK-ERK1/2 pathway, which was likely the result of a double positive impact on *ACOX2* and *CYP7A1* expression. As has been previously described for the transcriptional control of *CYP7A1*, the final effector accounting for ERK-mediated FGF19-induced *CYP7A1* and *ACOX2* repression remains unknown [56].

Deregulation of the BA metabolism and failure of the FXR signaling pathway have already been described in inflammatory liver diseases, such as NASH [65]. In mice, inflammation-activated signaling cascades have been suggested to play a predominant role (over FXR/SHP and FXR/FGF15 pathways) in regulating BA biosynthesis under inflammatory conditions [66].

IL1β, which down-regulates *CYP7A1* and *ACOX2*, but not *BAAT*, has been reported to inhibit the expression and activity of transporters involved in BA enterohepatic circulation such as NTCP, OATP1B1, OATP1B3, MRP2, BCRP and BSEP, which may participate in reducing bile flow and contribute to the development of cholestasis during inflammation [19]. The different efficacy of BA transporters in maintaining conjugated and unconjugated BAs in the enterohepatic circulation might be involved in the accumulation of the former over the latter in situations of inflammation-associated down-regulation of carrier proteins [2]. In addition, cytokine-induced down-regulation can slow down the rate of BA synthesis (CYP7A1/ACOX2) and conjugation (BAAT).

OSM, which belongs to the IL6 cytokine family, plays a crucial role in the maturation of fetal hepatocytes [67]. OSM is up-regulated in cirrhosis, where it has been associated with enhanced collagen production by stellate cells [68]. Upon binding to its receptor (OSMR) in hepatocytes, OSM also inhibits the expression of BA transporters [69]. We have observed that OSM is a particularly strong controller of both *ACOX2* and *BAAT* expression. Moreover, OSM mRNA is increased in the liver of patients with inflammation, which is in line with the results described by other authors [70]. More precisely, regarding *BAAT* expression, our results indicate that OSM is a strong repressor, while TNFα and IL1β lack this ability. OSM-induced *BAAT* down-regulation was dependent on JAK, PKCδ and ERK1/2 activation, which has been described to recruit linker proteins that intracellularly propagate the signal [71]. This is consistent with the fact that OSM interaction with OSMR produces a particularly prominent activation of the ERK pathway [72]. The regulation of *ACOX2* expression seems to share these control pathways with that of *BAAT*, whereas only in the case of *ACOX2* might the MEK5-mediated pathway be involved. Interestingly, in contrast with the commonly transient effect of most cytokines, due to inhibitory feedback loops, the effect of OSM on *ACOX2* and *BAAT* expression was markedly persistent, which could be partly due to the ability of OSM to remain anchored to the extracellular matrix in an active state [73]. This finding is relevant in the context of liver fibrosis, which is favored by OSM [68], because this cytokine may play a persistent role in keeping the activation of inflammatory and fibrogenic signals for a longer time than other cytokines, such as TNFα, IL1β and IL6 [74].

In conclusion, hepatic BA side chain shortening and conjugation are inhibited by inflammatory effectors. Moreover, our results support the existence of an interconnection between HNF4α and *ACOX2/BAAT* expression during NASH development. Under these circumstances, the transcriptional control of *ACOX2/BAAT* does not seem to be parallel to that of *CYP7A1*. Interestingly, although ACOX2 and BAAT expression is affected by inflammatory cytokines in NASH, other mechanisms involved in the overall BA homeostasis, which need to be unraveled, overcome the impact of these changes on the serum BA profile in these patients. Our results indicate that the reduction in ACOX2 enzymatic activity does not limit the generation of mature BAs, which only occurs when ACOX2 activity is abolished due to genetic defects, resulting in the accumulation of C27-BAs [29]. In NASH patients, the lack of *CYP7A1* repression, together with the increased levels of C4 in serum, suggests that there is no appropriate activation of the intestinal FXR/FGF19 axis, perhaps due to an insufficient level of BAs in intestinal cells. This, together with the enhanced concentration of BAs in serum, mainly conjugated forms despite *BAAT* down-regulation, is consistent with the down-regulation of transporters involved in BA secretion into bile and re-uptake by the ileum. If this limitation favors the conservation of conjugated over non-conjugated BAs, it could explain our results. Moreover, decreased BA deconjugation associated with dysbiosis present in NASH patients can also be involved. These unanswered questions can form the basis for further investigations in this field.

## Figures and Tables

**Figure 1 cells-11-03983-f001:**
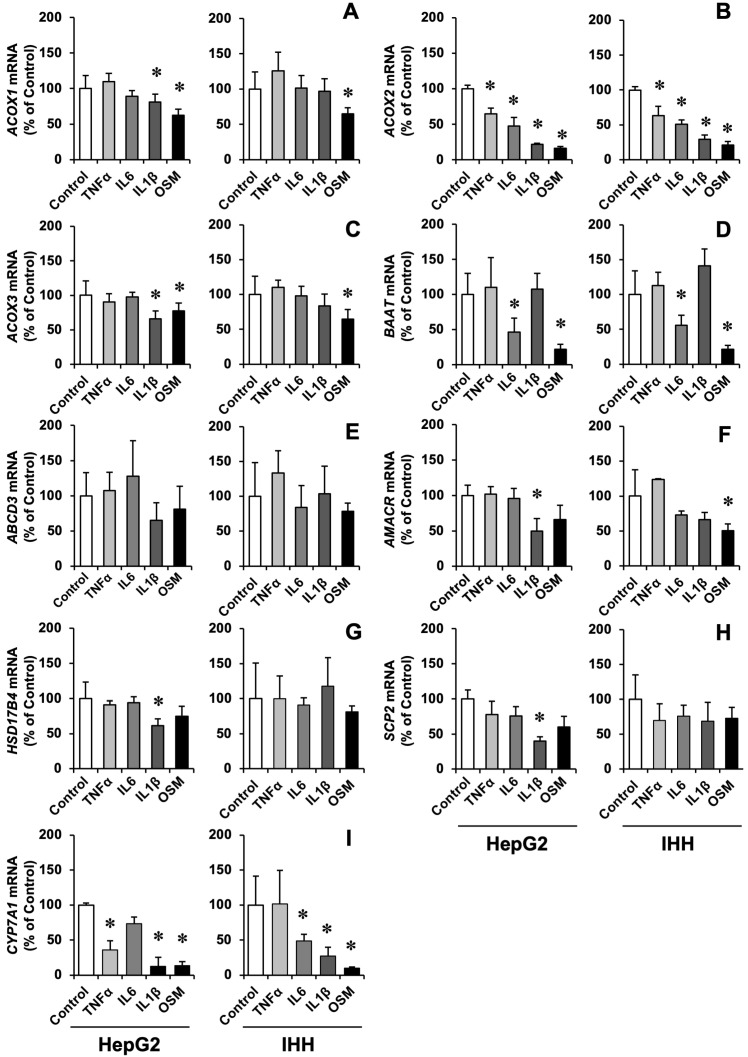
Relative mRNA levels, determined by RT-qPCR, of the enzymes involved in peroxisomal metabolism and transport ACOX1 (**A**), ACOX2 (**B**), ACOX3 (**C**), BAAT (**D**), ABCD3 (**E**), AMACR (**F**), HSD17B4 (**G**), and SCP2 (**H**), and in BA biosynthesis CYP7A1 (**I**) in HepG2 and IHH cells in HepG2 and IHH cells, which were incubated with 50 ng/mL tumor necrosis factor α (TNFα), 50 ng/mL interleukin 6 (IL6), 10 ng/mL interleukin 1β (IL1β) or 25 ng/mL oncostatin M (OSM) for 6 h. Results (mean ± SD) from at least three determinations carried out in three different cultures are expressed as a percentage of values found in untreated cells (Control). *, *p* < 0.05, as compared with Control by paired *t*-test.

**Figure 2 cells-11-03983-f002:**
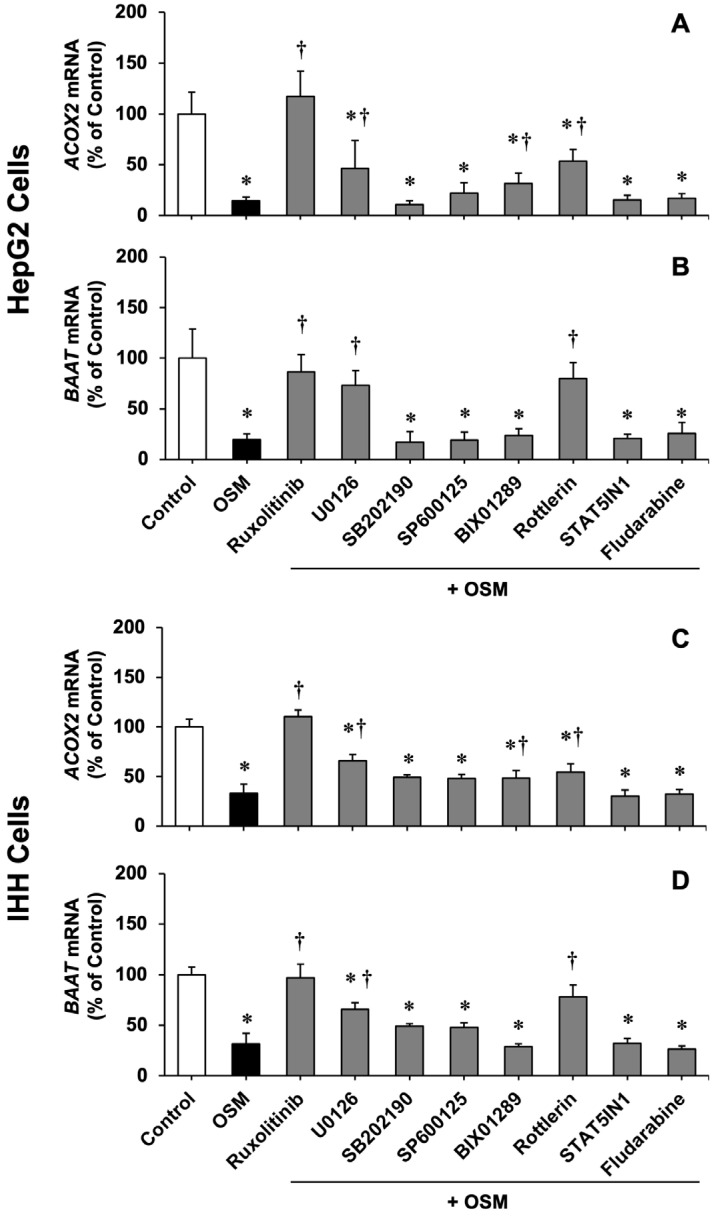
Relative mRNA levels, determined by RT-qPCR, of *ACOX2* (**A**,**C**) and *BAAT* (**B**,**D**) in HepG2 (**A**,**B**) and IHH (**C**,**D**) cells, which were incubated with 25 ng/mL oncostatin M (OSM) alone or with 0.5 µM ruxolitinib (JAK inhibitor), 5 µM U0126 (MEK1/2 inhibitor), 5 µM SB202190 (p38 MAPK inhibitor), 5 µM SP600125 (JNKs inhibitor), 10 µM BIX01289 (MEK5 inhibitor), 10 µM rottlerin (PKCδ inhibitor), 50 µM STAT5IN1 (STAT5 inhibitor), or 10 µM fludarabine (STAT1 inhibitor) for 6 h. Results (mean ± SD) from six determinations carried out in three different cultures are expressed as a percentage of values found in untreated cells (Control). *, *p* < 0.05, as compared with Control by paired *t*-test. †, *p* < 0.05, as compared with cells treated only with OSM by paired *t*-test.

**Figure 3 cells-11-03983-f003:**
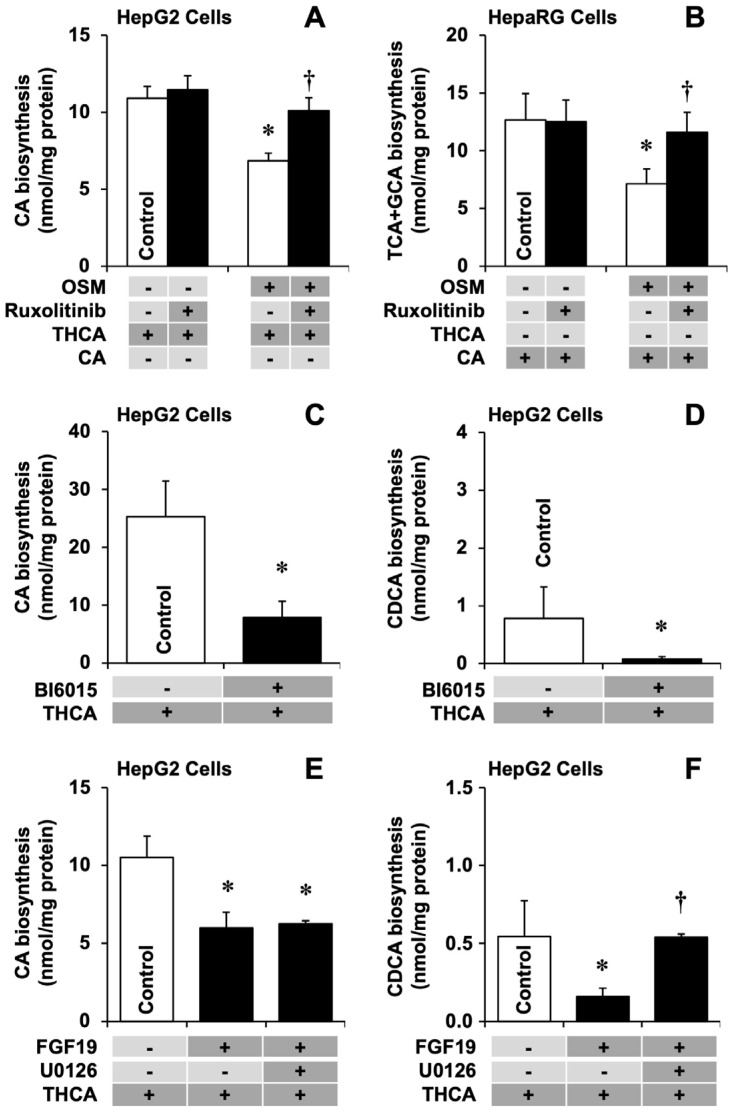
Effect of oncostatin M (OSM, 25 ng/mL) in the absence or presence of the JAK inhibitor ruxolitinib (0.5 µM) on cholic acid (CA) biosynthesis from trihydroxycholestanoic acid (THCA, 10 µM) by HepG2 cells (**A**) and taurocholic acid (TCA) plus glycocholic acid (GCA) biosynthesis from CA (5 µM) by differentiated HepaRG cells (**B**). Effect of the HNF4α inhibitor BI6015 (5 µM) on the ability of HepG2 cells to biotransform exogenous THCA (10 µM) into CA (**C**) and generate chenodeoxycholic acid (CDCA) from endogenous dihydroxycholestanoic acid (**D**). Effect of recombinant human FGF19 (50 ng/mL) alone or in the presence of the MEK1/2 inhibitor U0126 (5 µM) on the ability of HepG2 cells to biotransform exogenous THCA (10 µM) into CA (**E**) and to generate CDCA from endogenous dihydroxycholestanoic acid (**F**). After the initial 6 h in the absence of BA substrates, THCA or CA were added and maintained for the following 48 h. BA content in the culture medium and cell lysates was then analyzed. Results (mean ± SD) from six determinations that were carried out in three different cultures. BA species were measured by HPLC-MS/MS. *, *p* < 0.05, as compared to Control by paired *t*-test. †, *p* < 0.05, as compared with cells incubated in the absence of the inhibitor used in each group by paired *t*-test.

**Figure 4 cells-11-03983-f004:**
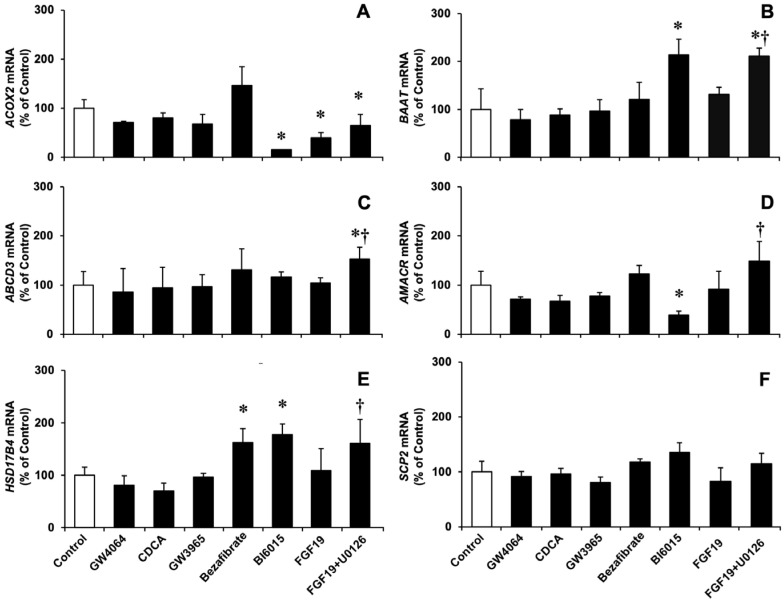
Relative mRNA levels, determined by RT-qPCR, of *ACOX2* (**A**), *BAAT* (**B**), *ABCD3* (**C**), *AMACR* (**D**), *HSD17B4* (**E**), and *SCP2* (**F**) in HepG2 cells, which were incubated for 24 h with the FXR agonists GW4064 (1 µM) or chenodeoxycholic acid (CDCA, 50 µM), the LXR agonist GW3965 (1 µM), the PPAR agonist bezafibrate (80 µM), the HNF4α inhibitor BI6015 (10 µM), or human recombinant FGF19 (50 ng/mL) alone or combined with the MEK1/2 inhibitor U0126 (5 µM). Results are expressed as a percentage of the values determined in untreated cells (Control) and represent the mean ± SD from at least three determinations that were carried out in three different cultures. *, *p* < 0.05, as compared to Control by paired *t*-test. †, *p* < 0.05, as compared with cells incubated with FGF19 alone by paired *t*-test.

**Figure 5 cells-11-03983-f005:**
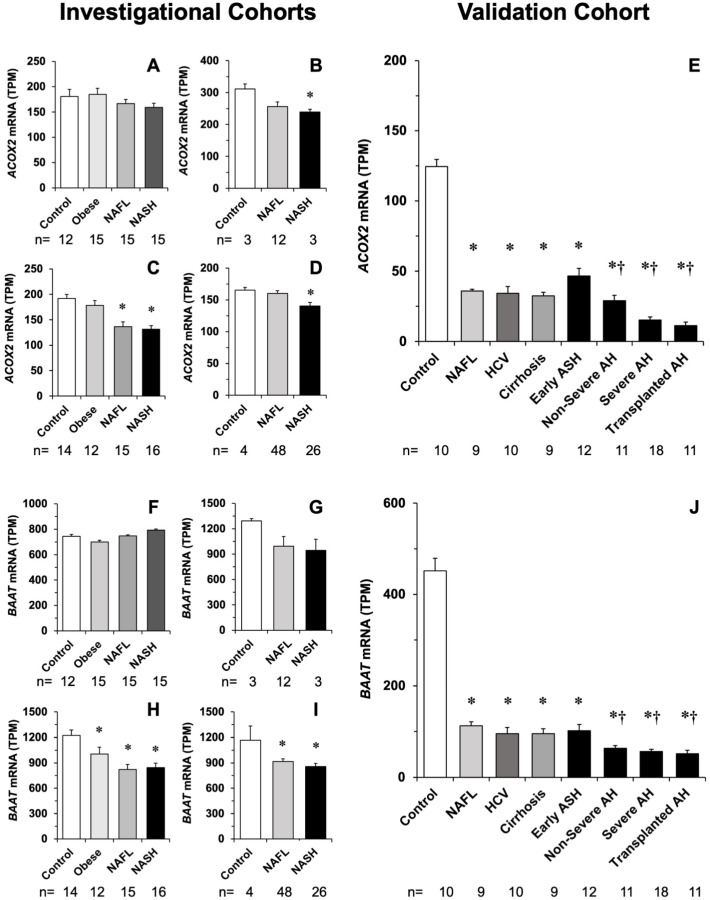
Abundance of *ACOX2* and *BAAT* mRNA (TPM) in the liver of patients belonging to four investigational cohorts, whose information is available in databases GSE105127 (**A**,**F**), GSE115193 (**B**,**G**), GSE126848 (**C**,**H**) and GSE130970 (**D**,**I**). Determination of mRNA abundance by similar methods in a validation cohort with different liver diseases (**E**,**J**). ANOVA, followed by the Bonferroni method of multiple-range testing was used for statistical analysis. *, *p* < 0.05 as compared with Control; †, *p* < 0.05 as compared with early ASH.

**Figure 6 cells-11-03983-f006:**
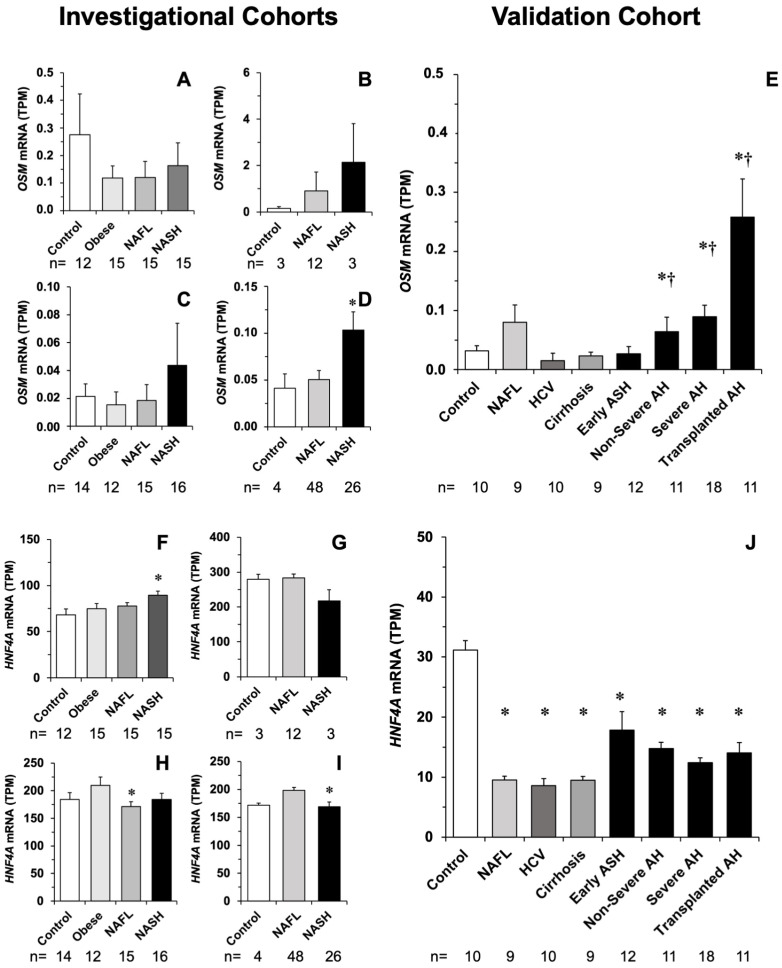
Abundance of *OSM* and *HNF4α* mRNA (TPM) in the liver of patients belonging to four investigational cohorts, whose information is available in databases GSE105127 (**A**,**F**), GSE115193 (**B**,**G**), GSE126848 (**C**,**H**) and GSE130970 (**D**,**I**). Determination of mRNA abundance by similar methods in a validation cohort with different liver diseases (**E**,**J**). ANOVA, followed by the Bonferroni method of multiple-range testing was used for statistical analysis. *, *p* < 0.05 as compared with Control; †, *p* < 0.05 as compared with Early ASH.

**Figure 7 cells-11-03983-f007:**
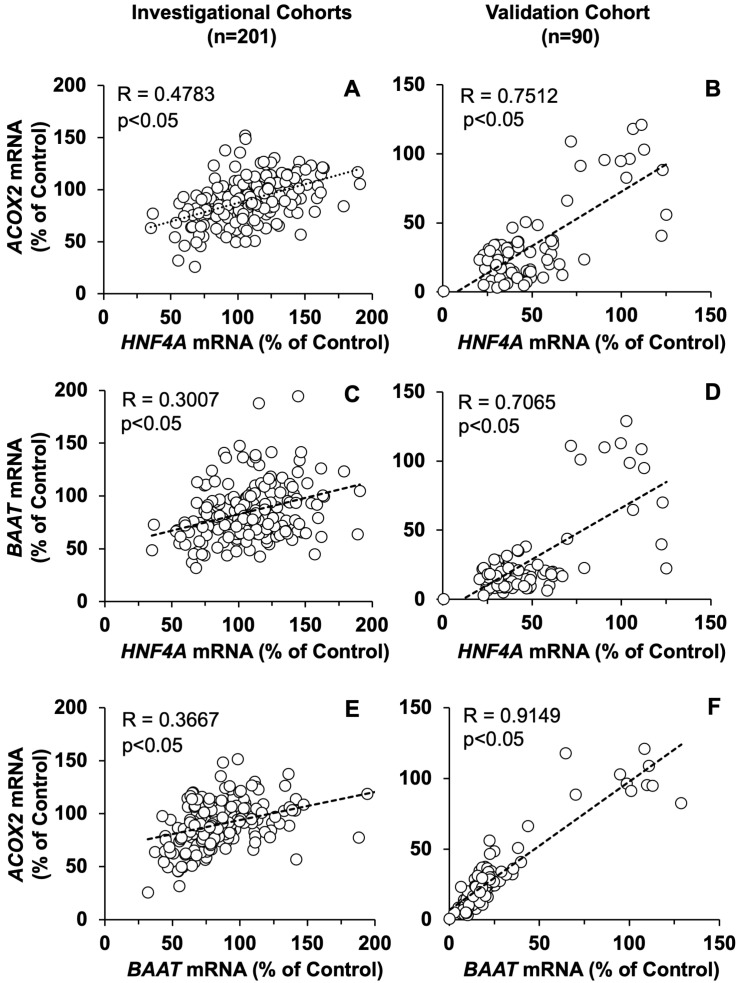
Inter-relationship among the mRNA expression levels of *ACOX2, BAAT,* and *HNF4α* in the liver of patients belonging to the investigational (**A**,**C**,**E**) or validation (**B**,**D**,**F**) cohorts. To correlate values obtained from different genes in different cohorts, the percentage changes in the expression of each gene versus the average value of mRNA abundance in the control group of the same dataset were used in the plot. These data were analyzed to calculate the statistical significance of the Spearman’s rank correlation coefficient (Wessa, P. 2022, Free Statistics Software, Office for Research Development and Education, version 1.2.1, URL https://www.wessa.net; accessed on 20 May 2022).

**Figure 8 cells-11-03983-f008:**
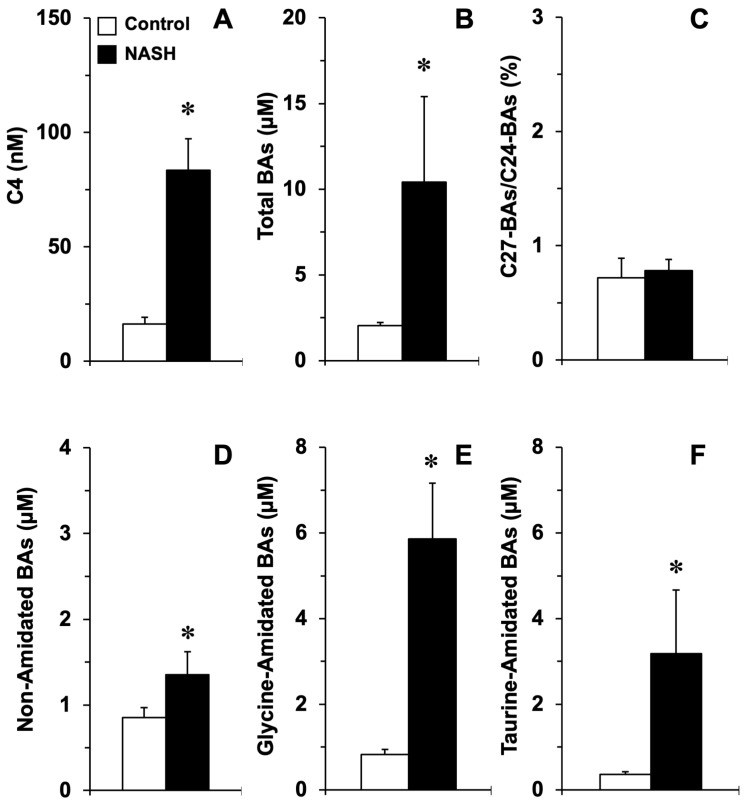
Concentrations of the bile acid (BA) precursor 7α-hydroxy-4-cholesten-3-one (C4) (**A**), total BAs (**B**), the proportion of C27-BAs (**C**), unconjugated BAs (**D**), and BAs amidated with glycine (**E**), or taurine (**F**) in the serum of healthy subjects (Control, *n* = 27) and patients with non-alcoholic steatohepatitis (NASH grade F3-F4; *n* = 29). Serum BA species and C4 were measured by HPLC-MS/MS. *, *p* < 0.05, as compared to Control by the Student’s unpaired *t*-test.

**Figure 9 cells-11-03983-f009:**
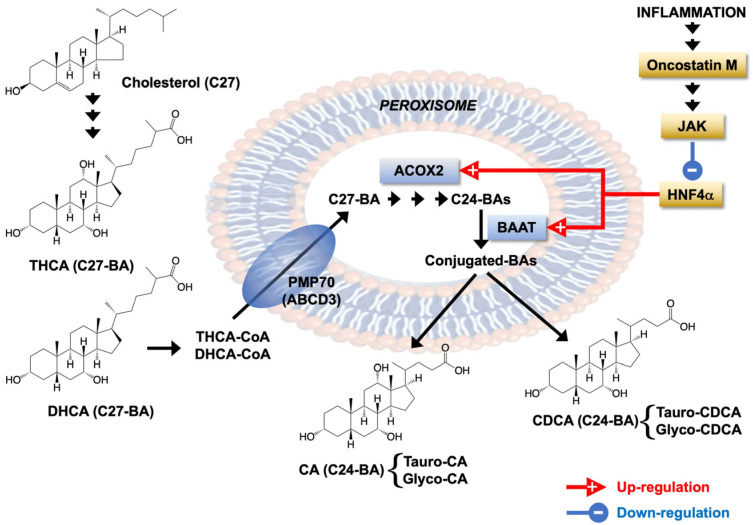
Schematic representation of the effect of inflammation-associated cytokines oncostatin M, IL6, IL1β and TNFα on the expression of peroxisomal enzymes acyl-CoA oxidase 2 (ACOX2) and BA-CoA:amino acid N-acyltransferase (BAAT) involved in bile acid (BA) side chain shortening and conjugation with taurine or glycine, respectively. PMP70, 70-kDa peroxisomal membrane protein or ABCD3 transporter; CA, cholic acid; CDCA, chenodeoxycholic acid; DHCA, dihydroxycholestanoic acid; THCA, trihydroxycholestanoic acid.

## Data Availability

Datasets used for in silico studies are available at https://www.ncbi.nlm.nih.gov/geo/ (accessed on 20 May 2022) under the indicated GEO accession references.

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
