# Peer review of "Impact of Liver Inflammation on Bile Acid Side Chain Shortening and Amidation"

_cells, 2022, doi:10.3390/cells11243983_

Round 1

Reviewer 1 Report

In this MS, Alonso-Peña and co-authors tested the effect of different cytokines on the expression of genes encoding proteins involved in the BA biosynthesis, BA shortening and conjugation. The authors observed an obvious inhibitory effect on the expression of ACOX2 and BAAT by oncostatin, and differential effects of other cytokines on the various genes. In addition, the authors also tested the effects of different signaling inhibitors alone or in conjunction with oncostatin on the expression ACOX2 and BAAT with the purpose to determine the signaling pathways involved in the inhibitory effect of oncostatin. Next, the authors tried to connect their observations in the cultured cells to the human patients. As a whole, this work has been well designed and has thoroughly tested possible links between inflammation with BA shortening and conjugation. The results obtained are meaningful and the conclusions made are justifiable.

I have only one suggestion: the authors need to provide a diagram showing the BA metabolic processes (synthesis, shortening and conjugation) by including the enzymes and regulators involved, and the tissue and subcellular location of these biochemical process taking place. This will be more informative and make this work easier for the audience to follow.       

Author Response

In this MS, Alonso-Peña and co-authors tested the effect of different cytokines on the expression of genes encoding proteins involved in the BA biosynthesis, BA shortening and conjugation. The authors observed an obvious inhibitory effect on the expression of ACOX2 and BAAT by oncostatin, and differential effects of other cytokines on the various genes. In addition, the authors also tested the effects of different signaling inhibitors alone or in conjunction with oncostatin on the expression ACOX2 and BAAT with the purpose to determine the signaling pathways involved in the inhibitory effect of oncostatin. Next, the authors tried to connect their observations in the cultured cells to the human patients. As a whole, this work has been well designed and has thoroughly tested possible links between inflammation with BA shortening and conjugation. The results obtained are meaningful and the conclusions made are justifiable.

I have only one suggestion: the authors need to provide a diagram showing the BA metabolic processes (synthesis, shortening and conjugation) by including the enzymes and regulators involved, and the tissue and subcellular location of these biochemical process taking place. This will be more informative and make this work easier for the audience to follow.

RESPONSE: We thank the reviewer for the positive evaluation of our study. Following the reviewer’s recommendation, a graphical abstract and a new figure 9 including all items suggested has been generated and submitted with the new version of the manuscript.

Reviewer 2 Report

Alonso-Pena et al combine several approaches in this research paper to investigate the impact of inflammatory processes on the expression of peroxisomal enzymes involved in the shortening and conjugation of BA side chain and their potential relationship with the altered BA homeostasis described in these patients. The paper is well written and well presented and explores a relevant field in NAFLD.

COMMENTS:

1.     I am not sure if I am missing files or if the website does not include all supplementary information files. If not the case, I would strongly recommend authors to revise and improve their presentation and include appropriate figure legends and loading controls where missing.

2.     I was wondering if it would be possible to include more data regarding the validation cohort used in figure 6 (E and J) and to perform univariate and multivariate testing against other relevant clinical features besides disease aetiology (e.g. histological/biochemical signs of hepatocellular damage and cholestatic damage, inflammatory markers, fibrosis (histology, FIB-4)).  I think the paper will benefit from integrating this type of analyses without much more effort and help concreting to which features of the disease are the expression of these genes linked to.

3.     Please include the squared R in figure 7.

Author Response

Alonso-Pena et al combine several approaches in this research paper to investigate the impact of inflammatory processes on the expression of peroxisomal enzymes involved in the shortening and conjugation of BA side chain and their potential relationship with the altered BA homeostasis described in these patients. The paper is well written and well presented and explores a relevant field in NAFLD.

  1. I am not sure if I am missing files or if the website does not include all supplementary information files. If not the case, I would strongly recommend authors to revise and improve their presentation and include appropriate figure legends and loading controls where missing.

RESPONSE: This comment has been coincident with that by another reviewer. However, Supplementary information (six figures and two tables, including figure legends) is contained in the compressed (.zip) folder named “manuscript.zip” on the website of the journal. We have confirmed that this can be downloaded and expanded. Should it be a technical problem?

  1. I was wondering if it would be possible to include more data regarding the validation cohort used in figure 6 (E and J) and to perform univariate and multivariate testing against other relevant clinical features besides disease aetiology (e.g. histological/biochemical signs of hepatocellular damage and cholestatic damage, inflammatory markers, fibrosis (histology, FIB-4)). I think the paper will benefit from integrating this type of analyses without much more effort and help concreting to which features of the disease are the expression of these genes linked to.

RESPONSE: In the current study, we focused our interest on the signaling role of effectors of inflammation. Our group is currently trying to elucidate the relationship between changes in bile acid metabolism and the severity of liver inflammation (e.g., degree of fibrosis). Performing the proposed multivariate in silico study on the analyzed data set would be complex and not always possible and will change the whole orientation of the paper. However, we believe that the reviewer’s recommendation is very appealing. Accordingly, we will consider it for further steps in this line of research.

  1. Please include the squared R in figure 7.

RESPONSE: Following the reviewer’s recommendation, the R value has been included in Figure 7, but also in Supplementary Figure 6, which also include similar correlation curves. 

Reviewer 3 Report

The work of Marta Alonso-Peña and her colleagues postulates altered bile salt chain shortening and conjugation by inflammatory processes in liver diseases such as NASH via oncostatin. The expected effect, decrease in conjugated bile salts in NASH, however, is not detectable when bile acids have been analyzed in patients. The authors ultimately suspect other mechanisms that nevertheless elevate bile salt concentration in inflammatory liver diseases. The authors should address the following aspects to further improve their study:

1. A graphic abstract, which would facilitate the understanding of this study, is unfortunately missing.

2. Abstract: Explain abbreviations such as ACOX2 and AMACR etc. to the reader the first time they are used.

3. Line 74: Orphan receptors are receptors with related structure to other identified receptors, but their ligand has not yet been identified. HNF4a is a transcription factor with known effects in maintaining liver architecture and function.

4. Lines 85-86: Abbreviations of TNFa, IL1b and bile acid transporters must be explained.

5. Line 89: It should be mentioned here that phosphorylation inhibits HNF4a.

6. Lines 93-100: A scheme showing the bile acid synthesis and processes that are involved in bile acid conjugation in the peroxisomes would be helpful for proper understanding.

7. Line 127: Can the authors provide an official case number for ethical approval of the study.

8. Line 160 and line 177: cm²

9. Line 191: Abbreviation HPRT

10. Lines 187 and 196 etc. The supplement was not uploaded by the authors or was not accessible to me.

11. Lines 206 and 221: Were the age and gender of the patients considered?

12. Line 254: “by (several) inflammatory cytokines” Be more specific and provide a detailed list.

13. Line 272: “(several) pathways were manipulated” Please be more specific.

14. Line 278: “In contrast” was used twice.

15. Lines 280-285: So, OSM is not acting via JAK/STAT3 signaling. Which signaling pathway then led to the activation of ERK1/2 by OSM?

16. Figure 3A/B: This figure is difficult to understand. Why were two controls shown? What is meant by TCA+GCA? Is the sum of both bile acids shown? Since glycine conjugation is prevalent in human hepatocytes, it might be that GCA is more relevant than TCA.

17. Lines 384-406: The work would benefit from also examining the protein levels of HNF4a, ACOX2 and BAAT in liver tissue from patients with NASH. The mRNA has only limited significance.

18. Line 433: There are already expression analyses of mice with a hepatocyte-specific knockout of HNF4a (see PMID:22241473). In these publicly available datasets, BAAT and ACOX2 should also be altered. This would strengthen the validity of the study.

19. Figure 8: The authors have postulated a defect in bile salt biosynthesis based on the expression of HNF4a as well as ACOX2 and BAAT, but this cannot be observed based on the bile salt concentrations in the blood of patients with NASH. Have I understood this correctly? If so, the significance of the study should be better explained to the reader.

Author Response

The work of Marta Alonso-Peña and her colleagues postulates altered bile salt chain shortening and conjugation by inflammatory processes in liver diseases such as NASH via oncostatin. The expected effect, decrease in conjugated bile salts in NASH, however, is not detectable when bile acids have been analyzed in patients. The authors ultimately suspect other mechanisms that nevertheless elevate bile salt concentration in inflammatory liver diseases. The authors should address the following aspects to further improve their study:

  1. A graphic abstract, which would facilitate the understanding of this study, is unfortunately missing.

RESPONSE: We thank the reviewer for the positive evaluation of our study. Following the reviewer’s recommendation, a graphical abstract including all items suggested has been generated and submitted with the new version of the manuscript.

  1. Abstract: Explain abbreviations such as ACOX2 and AMACR etc. to the reader the first time they are used.

RESPONSE: Abbreviations for the main enzymes (ACOX2, AMACR, BAAT, and HSD17B4) involved in bile acid metabolism have been defined in the Abstract, as suggested.

  1. Line 74: Orphan receptors are receptors with related structure to other identified receptors, but their ligand has not yet been identified. HNF4a is a transcription factor with known effects in maintaining liver architecture and function.

RESPONSE: This mistake has been corrected.

  1. Lines 85-86: Abbreviations of TNFa, IL1b and bile acid transporters must be explained.

RESPONSE: Most abbreviations have been defined.

  1. Line 89: It should be mentioned here that phosphorylation inhibits HNF4a.

RESPONSE: A sentence in this respect has been added together with a new reference to support this statement.

  1. Lines 93-100: A scheme showing the bile acid synthesis and processes that are involved in bile acid conjugation in the peroxisomes would be helpful for proper understanding.

RESPONSE: A new figure 9 has been added to summarize the metabolic steps and cytokine-mediated changes involved in the issues addressed in the study.

  1. Line 127: Can the authors provide an official case number for ethical approval of the study.

RESPONSE: The section “2.1. Ethics” has been improved as required.

  1. Line 160 and line 177: cm²

RESPONSE: This mistake has been corrected

  1. Line 191: Abbreviation HPRT

RESPONSE: The HPRT abbreviation has now been defined in the text.

  1. Lines 187 and 196 etc. The supplement was not uploaded by the authors or was not accessible to me.

RESPONSE: This comment has been coincident with that by another reviewer. However, Supplementary information (six figures and two tables, including figure legends) is contained in the compressed (.zip) folder named “manuscript.zip” on the website of the journal. We have confirmed that this can be downloaded and expanded. Should it be a technical problem?

  1. Lines 206 and 221: Were the age and gender of the patients considered?

RESPONSE: Data from all patients were analyzed without any stratification. Although this could be potentially interesting for some aspects of this study, where changes have been found in publicly available datasets, in some cases, the number of patients was very low, and in others, information regarding age and gender was not available, precluding this type of analysis.

  1. Line 254: “by (several) inflammatory cytokines” Be more specific and provide a detailed list.

RESPONSE: This sentence has been improved by including the list of cytokines concerned, as suggested.

  1. Line 272: “(several) pathways were manipulated” Please be more specific.

RESPONSE: This sentence has been rewritten in a more precise way.

  1. Line 278: “In contrast” was used twice.

RESPONSE: This sentence has been modified to remove the repetition of words.

  1. Lines 280-285: So, OSM is not acting via JAK/STAT3 signaling. Which signaling pathway then led to the activation of ERK1/2 by OSM?

RESPONSE: The reviewer is right. Our results suggest that OSM seems to act on ACOX2 expression in a STAT-independent manner, but probably through the sequence JAK / SHC+Grb2 / Ras / Raf / MEK / ERK (https://www.rndsystems.com/pathways/oncostatin-m-signaling-pathways). A sentence to clarify this point has been added together with two new references to support this hypothesis.

  1. Figure 3A/B: This figure is difficult to understand. Why were two controls shown? What is meant by TCA+GCA? Is the sum of both bile acids shown? Since glycine conjugation is prevalent in human hepatocytes, it might be that GCA is more relevant than TCA.

RESPONSE: The reviewer is right. There was a mistake in the labeling of white bars in Figures 3A and 3B. The “Control” label is appropriate only in the left white bar of each graph. This mistake has been corrected.

Regarding conjugation, the reviewer is also right. TCA+GCA refers to the sum of the amounts of both conjugated forms of CA measured when CA was added to the culture. Despite the prevalent BA conjugation with glycine in humans in vivo, in all these in vitro experiments carried out with this human cell line, TCA accounted for most (>95%) conjugated BAs (TCA+GCA) (data not shown). This has been commented on in the text.

  1. Lines 384-406: The work would benefit from also examining the protein levels of HNF4a, ACOX2 and BAAT in liver tissue from patients with NASH. The mRNA has only limited significance.

RESPONSE: We agree on the limited information supplied by determining mRNA levels. Unfortunately, gene expression data come from RNAseq analysis carried out elsewhere. We do not have access to tissue samples analyzed in this study. As our results recommend carrying out a separate study on a different set of patients, where enough amount of tissue could be obtained to measure protein expression regarding the genes mentioned by the reviewer, we contemplate the possibility of doing so in further steps of this investigation.

  1. Line 433: There are already expression analyses of mice with a hepatocyte-specific knockout of HNF4a (see PMID:22241473). In these publicly available datasets, BAAT and ACOX2 should also be altered. This would strengthen the validity of the study.

RESPONSE: This was an excellent recommendation. We have carried out in silico analysis of the mentioned dataset, and, as speculated by the reviewer, significant changes have been found. This information, together with the appropriate reference, has been included in the new version of the Discussion section.

  1. Figure 8: The authors have postulated a defect in bile salt biosynthesis based on the expression of HNF4a as well as ACOX2 and BAAT, but this cannot be observed based on the bile salt concentrations in the blood of patients with NASH. Have I understood this correctly? If so, the significance of the study should be better explained to the reader.

RESPONSE: The reviewer is right both in pointing out the message and in the fact that this was not clearly stated in the previous version. We have modified the final paragraph to better explain the conclusions of our study.

Round 2

Reviewer 3 Report

The manuscript by Alonso-Peña and colleagues was significantly improved, although not all of the reviewer's suggestions could be implemented by the authors. There remain minor corrections that the authors should make:

A) The summary still contains abbreviations that are not directly explained.

B) The siRNA sequences for Stat3 and Mock are missing.

C) Supplementary Figure 4: "Deter(r)minations".

D) Supplemental Figure 6: "These data were the(m)n".

Author Response

Reviewer

The manuscript by Alonso-Peña and colleagues was significantly improved, although not all of the reviewer's suggestions could be implemented by the authors. There remain minor corrections that the authors should make:

  1. A) The summary still contains abbreviations that are not directly explained.

RESPONSE: All abbreviations included in the abstract have been defined.

  1. B) The siRNA sequences for Stat3 and Mock are missing.

RESPONSE: To carry out STAT3 silencing, a commercial kit supplied by Thermo Scientific Dharmacon was used. Unfortunately, the company does not hand out sequences. All information regarding this process has been added to the Materials and Methods section as follows:

“To carry out STAT3 silencing, a mix of four siRNA (On-TARGET PLUS siRNA human STAT3; ref: J-003544-7/8/9/10; 1:1:1:1) and a Mock control (On-TARGET PLUS control pool non-targeting; ref. D-001810-10-05) both supplied by Thermo Scientific Dharmacon (Horizon, Perkin Elmer, Colorado, US), were used for cell transfection with RNAiMAX lipofectamine (Invitrogen, Thermo Fisher Scientific).”

  1. C) Supplementary Figure 4: "Deter(r)minations".

RESPONSE: Several mistakes detected in figure legends have been corrected.

  1. D) Supplemental Figure 6: "These data were the(m)n"."

RESPONSE: Several mistakes detected in figure legends have been corrected.